# THERE IS NO TRADE-OFF: ENFORCING FAIRNESS CAN IMPROVE ACCURACY

## ABSTRACT

One of the main barriers to the broader adoption of algorithmic fairness in machine learning is the trade-off between fairness and performance of ML models: many practitioners are unwilling to sacrifice the performance of their ML model for fairness. In this paper, we show that this trade-off may not be necessary. If the algorithmic biases in an ML model are due to sampling biases in the training data, then enforcing algorithmic fairness may improve the performance of the ML model on unbiased test data. We study conditions under which enforcing algorithmic fairness helps practitioners learn the Bayes decision rule for (unbiased) test data from biased training data. We also demonstrate the practical implications of our theoretical results in real-world ML tasks.

## 1 INTRODUCTION

Machine learning (ML) models are routinely used to make or support consequential decisions in hiring, lending, sales *etc.*(Citron and Pasquale, 2014). This proliferation of ML models in decision making and decision support roles has led to concerns that ML models may inherit (or even exacerbate) social biases in the training data. For example, Pro-Publica's investigation of Northpointe (now Equivant)'s COMPAS recidivism prediction tool revealed racial biases against African-Americans (Angwin et al., 2016).

In response, the ML community has developed many rigorous definitions of algorithmic fairness, including calibration (Corbett-Davies and Goel, 2018), (statistical) parity (Feldman et al., 2014), equalized odds (Hardt et al., 2016), and individual fairness (Dwork et al., 2011). Researchers have also designed many algorithms for enforcing the definitions during training (Agarwal et al., 2018; Cotter et al., 2019; Yurochkin et al., 2020). Despite this flurry of work, algorithmic fairness practices remain uncommon in production.

We conjecture that the lack of broader adoption of algorithmic fairness practices is because there seems to be a trade-off between accuracy and fairness. Many algorithms that enforce fairness solve optimization problems that maximize how well the model fits the training data subject to fairness constraints. The trade-off arises because imposing fairness constraints usually leads to a model that fits the training data less well (compared to a model from maximizing goodness-of-fit without any extra constraints).

In practice, this trade-off may not be relevant because the training data may be biased. For example, a resume screening model may reject most female applicants for technical roles because women are historically underrepresented in STEM fields, so women are underrepresented in the training data. This is a form of sampling bias, and it causes the model to perform poorly at test time because women are better represented in STEM fields today. In this example, the trade-off is irrelevant because we are mostly concerned with out-of-distribution (OOD) performance of the model.

There are many other examples of algorithmic bias arising due to biases in the training data. As another example, the systemic racism in the US criminal justice system disproportionately affects African-Americans, leading to higher rates of arrest, conviction, and incarceration. It is no surprise that recidivism prediction instruments trained on such biased data is biased against African-Americans (Angwin et al., 2016). In 2014, then U.S. Attorney General Eric Holder warned that recidivism prediction instruments "may exacerbate unwarranted and unjust disparities that are already far too common in our criminal justice system and in our society".

In this paper, we study whether the common algorithmic fairness practice of enforcing equal accuracy on certain segments of the population improves the OOD performance of the model. Such algorithmic fairness practices are common enough that there are methods (Agarwal et al., 2018; 2019) and software (*e.g.* TensorFlow Constrained Optimization (**?**)) devoted to operationalizing them. This provides an alternative argument for broader adoption of algorithmic fairness practices. Instead of viewing fairness as an intrinsically desirable property of ML models, we show that enforcing fairness helps ML models overcome biases in the training data. Our main contributions are:

1. We decompose the bias in the training data into two parts: a recoverable part orthogonal to the fair constraint and a non-recoverable part. We also derive necessary and sufficient conditions under which enforcing fairness on the training data leads to the Bayes optimal model at test time (see Theorem 3.4).
2. We show that it is possible to completely overcome the recoverable part of the bias (hence its name) by enforcing an appropriate risk-based notion of algorithmic fairness. This is possible regardless of the magnitude of this part of the bias (see Corollary 3.5).
3. We specialize our results to recidivism prediction task and demonstrate the benefits of enforcing fairness empirically (see section 4).

## 2  PROBLEM SETUP

To keep things simple, we consider a standard classification setup. Our results generalize readily to other supervised learning problems (see Appendix C for details).

Let $\mathcal{X} \subset \mathbb{R}^d$ be the feature space, $\mathcal{Y}$ be the set of possible labels, and $\mathcal{A}$ be the set of possible values of the sensitive attribute. In this setup, training and test examples are tuples of the form $(X, A, Y) \in \mathcal{X} \times \mathcal{A} \times \mathcal{Y}$. If the ML task is predicting whether a borrower will default on a loan, then each training/test example corresponds to a loan. The features in $X$ may include the borrower's credit history, income level, and outstanding debts; the label $Y \in \{0, 1\}$ encodes whether the borrower defaulted on the loan; the sensitive attribute may be the borrower's gender or race.

Let $P^*$ and $\widetilde{P}$ be probability distributions on $\mathcal{X} \times \mathcal{A} \times \mathcal{Y}$. We consider $P^*$ as the unbiased distribution from which samples at test time come from and $\widetilde{P}$ as the biased distribution from which the training data comes from. Let $\mathcal{H} = \{h : \mathcal{X} \to \mathcal{Y}\}$ be a model class (*e.g.* neural nets with a particular architecture) and $\ell$ be a loss function. Our goal is to learn the unbiased Bayes decision rule

$$h^* \in \arg\min_{h \in \mathcal{H}} L^*(h) \triangleq \mathbb{E}^*\big[\ell(h(X), Y)\big], \tag{2.1}$$

where $\mathbb{E}^*$ denotes expectation with respect to $P^*$, using only the biased training data from $\widetilde{P}$. Without further assumptions on $P^*$, this goal is impossible. To facilitate our goal, we assume the unbiased Bayes decision rule is algorithmically fair in some sense and hope that enforcing the correct notion of fairness allows us to recover $h^*$ from $\widetilde{P}$. We shall elaborate on the allowable differences between $P^*$ and $\widetilde{P}$ in subsection 2.2.

### 2.1  RISK-BASED NOTIONS OF ALGORITHMIC FAIRNESS

In this paper, we study the efficacy of enforces risk-based notions of algorithmic fairness in overcoming bias in the training data. To fix ideas, we provide two examples of risk-based notions of algorithmic fairness.

The first notion of algorithmic fairness we consider is **risk parity (RP)**. This definition is motivated by the notion of *demographic parity (DP)* in classification. Recall DP requires the *output* of the ML model $h(X)$ to be independent of the sensitive attribute $A$: $h(X) \perp A$. RP imposes a similar condition on the *risk* of the ML model.

**Definition 2.1** (risk parity). *ML model $h$ satisfies risk parity with respect to data distribution $P$ if*

$$\mathbb{E}_P\big[\ell(h(X), Y) \mid A = a\big] = \mathbb{E}_P\big[\ell(h(X), Y) \mid A = a'\big] \text{ for all } a, a' \in \mathcal{A}.$$

RP is widely used in practice to measure algorithmic bias in ML models. For example, the US National Institute of Standards and Technology (NIST) tested facial recognition systems and found

that the systems misidentify blacks at rates 5 to 10 times higher than whites Simonite (2019). By comparing the error rates of the system on blacks and whites, NIST is implicitly adopting RP as its definition of algorithmic fairness.

The second notion of algorithmic fairness that we consider is **conditional risk parity (CRP)**. This definition is similar to the notion of *equalized odds (EO)* (Hardt et al., 2016) in classification. Recall EO requires the *output* of the ML model $h(X)$ to be independent of the sensitive attribute $A$ *conditioned on the label*: $h(X) \perp A \mid Y$. CRP imposes a similar condition on the *risk* of the ML model; *i.e.* the risk of the ML model must be independent of the sensitive attribute conditioned on the label.

**Definition 2.2** (conditional risk parity). *An ML model $h$ satisfies conditional risk parity with respect to data distribution $P$ and label $y$ if*

$$\mathbb{E}_P\big[\ell(h(X), Y) \mid A = a, Y = y\big] = \mathbb{E}_P\big[\ell(h(X), Y) \mid A = a', Y = y\big] \text{ for all } a, a' \in \mathcal{A}, y \in \mathcal{Y}.$$

*We say an ML model $h$ satisfies CRP with respect to $P$ (without mentioning a label value) iff it satisfies CRP with respect to $P$ and all label values $y \in \mathcal{Y}$.*

We observe that EO implies CRP because $\ell(h(X), y)$ is a function of $h(X)$ after conditioning on $A$ and $Y$. CRP is also closely related to *error rate balance* (Chouldechova, 2017) and *overall accuracy equality* (Berk et al., 2017) in classification.

As we shall see, both RP and CRP are instances of risk-based notions of algorithmic fairness. The general form of such a notion is

$$\mathbb{E}_P\big[\ell(h(X), Y) \mid A = a, V = v\big] = \mathbb{E}_P\big[\ell(h(X), Y) \mid A = a', V = v\big] \text{ for all } a, a' \in \mathcal{A}, v \in \mathcal{V}, \tag{2.2}$$

where $V$ is known as the **discriminative attribute** (Ritov et al., 2017). To keep things simple, we assume $V$ is finite-valued, but it is possible to generalize our results to risk-based notions of algorithmic fairness with more general $V$'s (see Appendix C). For RP, $V$ is a trivial random variable; for CRP, $V$ is $Y$. It is not hard to see that risk-based notions of algorithmic fairness are equivalent to linear constraints on the **risk profiles** of ML models:

$$R(h) \triangleq \big[\mathbb{E}_P\big[\ell(h(X), Y) \mid A = a, V = v\big]\big]_{a \in \mathcal{A}, v \in \mathcal{V}}$$

The general fairness constraint has the form $R(h) \in \mathcal{F}$, where $\mathcal{F}$ is a subspace. Figure 1 provides some examples of risk sets and when it is possible to recover Bayes' classifier. We wrap up this subsection by presenting general structure of risk profiles under RP and CRP constraints:

**Example 2.3** (risk parity). *Define $R^{\mathrm{RP}}(h) \in \mathbb{R}^{|\mathcal{A}|}$ as the vector whose entries are*

$$R_a^{\mathrm{RP}}(h) \triangleq \mathbb{E}_P\big[\ell(h(X), Y) \mid A = a\big].$$

*In terms of $R_a^{\mathrm{RP}}(h)$, RP with respect to $P$ implies that $R_a^{\mathrm{RP}}(h) = c\mathbf{1}$ for some constant $c \in \mathbb{R}$. This is a linear constraint: the set of risk profiles that satisfy the RP constraint is the subspace*

$$\mathcal{F}_{\mathrm{RP}} \triangleq \{R \in \mathbb{R}^{|\mathcal{A}|} \mid R = c\mathbf{1}, \mathbf{1} \in \mathbb{R}^{|\mathcal{A}|}, c \in \mathbb{R}\}.$$

**Example 2.4** (conditional risk parity). *Define $R^{\mathrm{CRP}}(h) \in \mathbb{R}^{|\mathcal{A}| \times |\mathcal{Y}|}$ as the matrix whose entries are*

$$R_{a,y}^{\mathrm{CRP}}(h) \triangleq \mathbb{E}\big[\ell(h(X), Y) \mid A = a, Y = y\big].$$

*In terms of $R_{a,y}^{\mathrm{CRP}}(h)$, CRP with respect to $P$ implies $R^{\mathrm{CRP}}(h) = \mathbf{1}\mathbf{u}^\top$. This is again a linear constraint: the set of risk profiles that satisfy the CRP constraint is the subspace*

$$\mathcal{F}_{\mathrm{CRP}} \triangleq \{R \in \mathbb{R}^{|\mathcal{A}| \times |\mathcal{Y}|} \mid R = \mathbf{1}\mathbf{u}^\top, \mathbf{1} \in \mathbb{R}^{|\mathcal{A}|}, \mathbf{u} \in \mathbb{R}^{|\mathcal{Y}|}\}.$$

## 2.2 BIAS IN THE TRAINING DATA

In this subsection, we describe the allowable differences between the unbiased distribution $P^*$ and the distribution of the (biased) training data $\widetilde{P}$. To relate the risk of ML models on the training data and at test time, we assume that the risk profiles of the models with respect to $P^*$ and the profiles with respect to $\widetilde{P}$ are identical:

$$\mathbb{E}^*\big[\ell(h(X), Y) \mid A = a, V = v\big] = \widetilde{\mathbb{E}}\big[\ell(h(X), Y) \mid A = a, V = v\big] \text{ for all } a \in \mathcal{A}, v \in \mathcal{V}. \tag{2.3}$$

This assumption is similar to the covariate shift and label shift assumptions in transfer learning. It is (slightly) less restrictive because it only requires the expected value (instead of all moments) of the loss to be identical in the training data and at test time. In fact, both covariate and label shift imply instances of (2.3) (with appropriate discriminative attributes).

We also note that this assumption is implicit in enforcing risk-based notions of algorithmic fairness. If the risk profiles are not identical in the training data and at test time, then enforce risk-based notions of algorithmic fairness during training is pointless because constraints on the risk profiles in the training data do not generalize at test time.

Finally, we note that the choice of discriminative attributes is crucial. The general problem of learning the (unbiased) Bayes decision rule from biased training data is impossible because the (biased) training data may be totally uninformative of the risks of ML models at test time. A good choice of discriminative attributes keeps the training data informative by ensuring the risk profiles are identical on the training data and at test time. Here are two examples of good discriminative attributes.

**Example 2.5** (label bias). *In binary classification, training data may suffer from **label bias**. This kind of bias arises when positive examples from disadvantaged groups are under-represented in the training data. Here is an example of a data generating process that suffers from label bias: (i) sample training examples $(X_i, Y_i, A_i)$ from $P^*$, (ii) discard training examples from the disadvantaged group ($A_i = 0$) with positive label ($Y_i = 1$) with probability $\beta$. This leads to*

$$\widetilde{P}(X, Y, A) \propto P^*(X, Y, A) \cdot (1 - (1 - \beta)\mathbf{1}\{A = 0, Y = 1\}).$$

*Because there are fewer positive examples from the disadvantaged group in the training data (compared to test data), this kind of bias causes the ML model to predict mostly negative outcomes for the disadvantaged group. In practice, this kind of bias may creep into the training data more subtly. For example, if human judgements is a crucial part of the data generating process, then implicit biases may lead to over-representation of negative examples from disadvantaged groups in the training data (Yeom and Tschantz, 2019).*

*For training data with label bias, a good choice of discriminative attribute is the label. This is because the training data is a filtered version of the data at test time, and the filtering process only depends on the label (and sensitive attribute). Thus the class conditionals at test time are preserved in the training data; i.e. $\widetilde{P}_{X|a,y} = P^*_{X|a,y}$ for all $a \in \mathcal{A}$, $y \in \mathcal{Y}$.*

**Example 2.6.** *In some applications, positive examples of the disadvantaged group are not missing at random; their missingness may depend on some factor(s). In such cases, it is possible to keep risk profiles identical in the training data and at test time including the factor(s) as discriminative attributes. For example, the missingness of female applicants for technical roles may depend on their qualifications. By including (some measures of) qualification in the discriminative attribute, the model allows the missingness of female applicants to vary depending on their qualifications.*

## 2.3 Enforcing Algorithmic Fairness During Training

Recall our goal is to learn the (unbiased) Bayes decision rule (2.1). Unfortunately, we cannot solve (the empirical version of) (2.1) because the training data is biased. Instead, we consider solving (the empirical version of)

$$\left\{ \begin{array}{ll} \min_{h \in \mathcal{H}} & \widetilde{\mathbb{E}}\big[\ell(h(X), Y)\big] \\ \text{subject to} & R(h) \in \mathcal{F} \end{array} \right\} \equiv \left\{ \begin{array}{ll} \min_{R \in \mathcal{R}} & \langle \widetilde{P}_{A,V}, R \rangle \\ \text{subject to} & R \in \mathcal{F} \end{array} \right\}, \tag{2.4}$$

where $\mathcal{R} \triangleq \{R(h) \mid h \in \mathcal{H}\}$ is the set of all possible risk profiles and $\widetilde{P}_{A,V} \in [0, 1]^{|\mathcal{A}| \times |\mathcal{V}|}$ is the distribution of $(A, V)$ (so $\langle \widetilde{P}_{A,V}, R \rangle = \widetilde{\mathbb{E}}_{A,V}\big[R_{A,V}\big]$). We hope that the fair constraint in (2.4) corrects the bias in the training data. As we shall see, this is possible if (i) the discriminative attribute is chosen such that the risk profiles on the training data and at test time are identical, and (ii) the bias is small in certain "directions".

Before moving on to the main result, we remark that there are efficient algorithms for solving (2.4). One popular algorithm is a reductions approach by Agarwal et al. (2018). At a high level, the algorithm solves a sequence of weighted classification problems in which the weights are chosen so that the resulting classifier satisfies the desired algorithmic fairness constraints. This algorithm outputs randomized classifiers, which justifies one of the subsequent assumptions on (2.4).

## 3 BENEFITS AND DRAWBACKS OF FAIR RISK MINIMIZATION

The main result provides necessary and sufficient conditions for recovering the unbiased Bayes' classifier with (2.4). Before stating the main result, we state and justify our assumptions.

**Assumption 3.1.** *The unconstrained risk minimizer on unbiased data is algorithmically fair; i.e.* $\arg\min_{R \in \mathcal{R}} \langle P^*, R \rangle \subseteq \mathcal{F}$.

This assumption is necessary. If the unbiased Bayes classifier is not algorithmically fair, then there is no hope for (2.4) to recover the unbiased Bayes classifier; there will always be a bias term. This assumption is also implicit in large swaths of the algorithmic fairness literature. For example, Buolamwini and Gebru (2018) and Yang et al. (2020) suggest collecting representative training data to improve the accuracy of computer vision systems on individuals from underrepresented demographic groups. This suggestion implicitly assumes the Bayes classifier on representative training data is algorithmically fair. We refer to section 5 for a brief discussion on relaxing this assumption.

At first blush, it is tempting to think that because the unbiased Bayes classifier satisfies a fairness constraint, then enforcing this constraint always increases accuracy. Unfortunately, this is not the case: *enforcing algorithmic fairness may harm OOD generalization, even if the Bayes classifier at test time is algorithmically fair*. Intuitively, the assumption that $R^*$ is fair is a constraint on $P^*$, $\mathcal{R}$, and $\mathcal{F}$; it imposes no constraints on $\widetilde{P}$. By picking $\widetilde{P}$ adversarially, it is possible to have

$$\langle \widetilde{P}, \widetilde{R} \rangle \leq \langle \widetilde{P}, \widetilde{R}_{\mathcal{F}} \rangle.$$

Such examples are not pathological, and we provide a graphical example in Appendix A.

**Assumption 3.2.** *The risk set $\mathcal{R}$ is convex.*

This assumption is innocuous because it is possible to convexify the risk set by considering randomized decision rules. A randomized decision rule is a distribution on the hypothesis class. To evaluate a randomized decision rule $H$, we sample a decision rule $h$ from $H$ and evaluate $h$. It is not hard to see that the risk profiles of randomized decision rules are convex combinations of the risk profiles of (non-randomized) decision rules, so including randomized decision rules convexifies the risk set.

**Assumption 3.3.** *The risk profiles of the models in $\mathcal{H}$ are identical with respect to $P^*$ and $\tilde{P}$, i.e.*

$$\mathbb{E}^* \left[ \ell(h(X), Y) \mid A = a, V = v \right] = \tilde{\mathbb{E}} \left[ \ell(h(X), Y) \mid A = a, V = v \right] \text{ for all } a \in \mathcal{A}, v \in \mathcal{V}.$$

This assumption is needed to keep the risk profiles on the (biased) training data informative for the (unbiased) test data. We refer to section 2.2 for a more comprehensive discussion of this assumption.

**Theorem 3.4.** *Under assumptions 3.1, 3.2 and 3.3 the fair risk minimization (2.4) obtains $h \in \mathcal{H}$ such that $R(h) = R^*$ if and only if*

$$\Pi_{\mathcal{F}}(P^*_{A,V} - \widetilde{P}_{A,V}) - P^*_{A,V} \in \mathcal{N}_{\mathcal{R}}(R^*) + \mathcal{F}^{\perp}. \tag{3.1}$$

*where $P^*_{A,V}$ (resp. $\widetilde{P}_{A,V}$) is the marginal of $P^*$ (resp. $\widetilde{P}$) with respect to $(A, V)$, $\mathcal{N}_{\mathcal{R}}(R^*)$ is the normal cone of $\mathcal{R}$ at $R^*$ and $\Pi_{\mathcal{F}}$ is the projection on the fair hyperplane.*

Theorem 3.4 characterizes the biases in the training data from which it is possible to totally recover by enforcing appropriate algorithmic fairness constraints. By totally recover from bias, we mean recovering the unbiased Bayes decision rule. To keep things simple, we stated our main result only for finite-valued discriminative attributes. Please see Appendix C for a more general version of Theorem 3.4 that applies to more general (including continuous-valued) discriminative attributes.

The main insight from Theorem 3.4 (and its counterpart for continuous discriminative attributes in Appendix C) is a decomposition of the training bias into two parts: a part orthogonal to the fair constraint and the remaining part in $\mathcal{N}_{\mathcal{R}}(R^*)$. Enforcing an appropriate risk-based notion of algorithmic fairness overcomes the first part of the bias. This occurs regardless of the magnitude of this part of the bias (see Corollary 3.5), and we see this later in our computational results.

The second part of training bias (the part in $\mathcal{N}_{\mathcal{R}}(R^*)$) represents the "natural" robustness of $R^*$ to changes in $P^*$: if $\widetilde{P}$ is in $\mathcal{N}_{\mathcal{R}}(R^*)$, then the unconstrained risk minimizer on training data remains $R^*$. The magnitude of the bias in this set cannot be too large, and enforcing algorithmic fairness constraints does not help overcome this part of the bias.

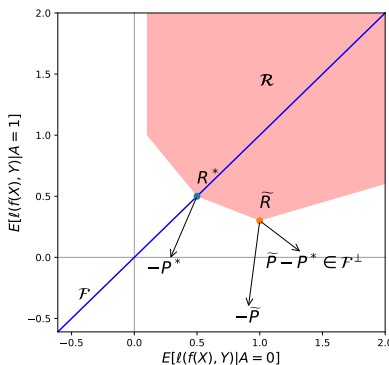

Figure 1: Total recovery from training bias by enforcing risk parity. In this simple example, the training bias $\widetilde{P} - P^*$ is always orthogonal to the risk parity constraint (blue line) because $\widetilde{P}$ and $P^*$ are probability distributions. Thus if the training bias does not affect the risk profiles (*i.e.* $\widetilde{P}$ satisfies Assumption 3.3), then enforcing risk parity allows us to totally overcome the training bias. Unfortunately, to show an example in which the risk decomposes into recoverable and non-recoverable parts, we need (at least) two more dimensions.

**Corollary 3.5.** *A sufficient condition for* (3.1) *is* $\widetilde{P}_{A,V} - P^*_{A,V} \in \mathcal{F}^\perp$.

*Proof of Corollary 3.5.* If $\widetilde{P}_{A,V} - P^*_{A,V} \in \mathcal{F}^\perp$, then $\Pi_\mathcal{F}(P^*_{A,V} - \widetilde{P}_{A,V}) = 0$, so we need to check that $-P^*_{A,V} \in \mathcal{N}_\mathcal{R}(R^*) + \mathcal{F}^\perp$. For any $R \in \mathcal{R}$,

$$\langle -P^*_{A,V}, R - R^* \rangle = \langle P^*_{A,V}, R^* - R \rangle \leq 0$$

as $R^*$ is the minimum value of $\langle P^*_{A,V}, R \rangle$ over $\mathcal{R}$. This shows $-P^*_{A,V} \in \mathcal{N}_\mathcal{R}(R^*)$ as desired. $\square$

Corollary 3.5 allows large differences between $\widetilde{P}_{A,V}$ and its unbiased counterpart $P^*_{A,V}$, as long as the differences are confined to $\mathcal{F}^\perp$. Intuitively, (2.4) enables practitioners to recover from large biases in $\mathcal{F}^\perp$ because the algorithmic fairness constraint "soaks up" any component of $\widetilde{P}_{A,V}$ in $\mathcal{F}^\perp$. We explore the implications of Corollary 3.5 for risk parity and CRP.

**Risk Parity:** For RP, $V$ is trivial random variable, hence $\widetilde{P}_A - P^*_A \in \mathcal{F}^\perp_{RP}$ means that it has mean 0. This is true for any $\widetilde{P}_A$ as $\langle P^*_A, 1 \rangle = \langle \widetilde{P}_A, 1 \rangle = 1$. Hence, the Bayes' classifier can be recovered under any perturbation. More specifically, recall the example of women historically underrepresented in STEM fields mentioned in the introduction. Such train data is biased in its gender representation which differs at test time where women are better represented. Classifiers trained on biased data with the risk Parity fairness constraint will generalize better at test time.

**Conditional risk parity:** In this case $V = Y$ and the condition $\widetilde{P}_{A,Y} - P^*_{A,Y} \in \mathcal{F}^\perp_{CRP}$ implies that the sum of each column of $\widetilde{P}_{A,Y} - P^*_{A,Y}$ must be 0. Hence, to recover the Bayes classifier under equalized odds fairness constraints, we are allowed to perturb $P^*_{A,Y}$ in such a way, that they have the same column sums: i.e. for any label, we are allowed to perturb the distribution of protected attributes for that label, but we have to keep the marginal distribution of the label to be same for both $\widetilde{P}_{A,Y}$ and $P^*_{A,Y}$. We investigate this scenario empirically in Section 4.

In practice, it is unlikely that the training bias is exactly orthogonal to the fair constraint, so Theorem 3.4 is a more general recovery result that characterizes conditions under which fair risk minimization recovers the Bayes classifier. For this to happen, the remaining part of the bias must be small enough. Theorem 3.4 provides a precise characterization of "small enough".

### 3.1 RELATED WORK

Most of the prior works on algorithmic fairness assume fairness is an intrinsically desirable property of an ML model, but this assumption is unrealistic in practice (Agarwal et al., 2018; Cotter et al., 2019; Yurochkin et al., 2020). There is a small but growing line of work on how enforcing fairness helps ML models recover from bias in the training data. Kleinberg and Raghavan (2018); Celis et al. (2020) consider strategies for correcting biases in hiring processes. They show that correcting the biases not only increases the fraction of successful applicants from the minority group but also boosts the quality of successful applicants. Dutta et al. (2019) study the accuracy-fairness trade-off in binary classification in terms of the separation of the classes within the protected groups. They

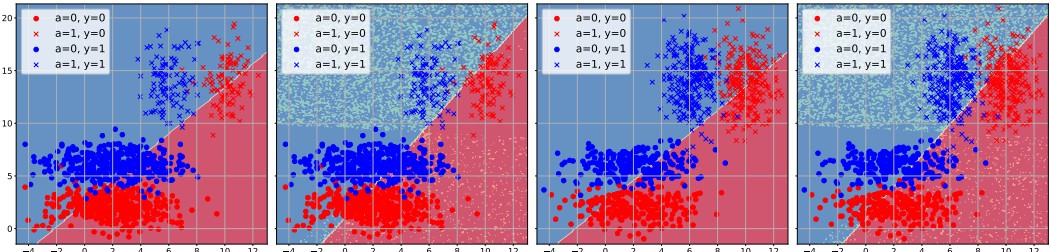

Figure 2: Decision heatmaps for (left) baseline on train data from $\widetilde{P}$; (center left) fair classifier on train data from $\widetilde{P}$; (center right) baseline on test data from $P^*$; (right) fair classifier on test data from $P^*$. Decision boundary of the fair classifier has larger slope better accounting for the group $a = 1$ underrepresented in the train data. Consequently its performance is better on the unbiased test data.

explain the accuracy-fairness trade-off in terms of this separation and propose a way of achieving fairness without compromising separation by collecting more features.

Blum and Stangl (2019) study how common group fairness criteria help binary classification models recover from bias in the training data. In particular, they show that the equal opportunity criteria (Hardt et al., 2016) recovers the Bayes classifier despite under-representation and labeling biases in the training data. Our results complement theirs. Instead of comparing the effects of enforcing various fairness criteria on training data with two types of biases, we characterize the types of biases that the fairness criteria help overcome. Our results also reveal the geometric underpinnings of the constants that arise in Blum and Stangl's results. Three other differences between our results and theirs are: (i) they only consider binary classification, while we consider all ML tasks that boil down to risk minimization, (ii) they allow some form of posterior drift (so the risk profiles of the models in $\mathcal{H}$ with respect to $P^*$ and $\widetilde{P}$ may differ in some ways), but only permit marginal drift in the label ($V = Y$), (iii) their conditions are sufficient for recovery of the fair Bayes decision rule (in their setting), while our conditions are also necessary (in our setting).

## 4 COMPUTATIONAL RESULTS

We verify the theoretical findings of the paper empirically. Our goal is to show that an algorithm trained with fairness constraints on the biased train data $\widetilde{P}$ achieves superior performance on the true data generating $P^*$ at test time in comparison to an algorithm trained without fairness considerations.

There are several algorithms in the literature that offer the functionality of empirical risk minimization subject to various fairness constraints, e.g. Cotter et al. (2019) and Agarwal et al. (2018). Any such algorithm will suffice to verify our theory. In our experiments we use Reductions fair classification algorithm (Agarwal et al., 2018) with logistic regression as the base classifier. For the fairness constraint we consider Equalized Odds (Hardt et al., 2016) (EO) — one of the major and more nuanced fairness definitions. We refer to Reductions algorithm trained with loose EO violation constraint as baseline and Reductions trained with tight EO violation constraint as fair classifier (please see Appendix D for additional details and supplementary material for the code).

**Simulations.** We first verify the implications of Corollary 3.5 using simulation studies. We follow the Conditional risk parity scenario from Section 3. Specifically, consider a binary classification problem with two protected groups, i.e. $Y \in \{0, 1\}$ and $A \in \{0, 1\}$. We set $P^*$ to have equal representation of protected groups conditioned on the label and biased data $\widetilde{P}$ to have one of the protected groups underrepresented. Specifically, let $p_{ay} = P_{A=a,Y=y}$, i.e. the $a, y$ indexed element of $P_{A,Y}$; $p_{ay} = 0.25 \; \forall a, y$ for $P^*$ and $p_{1y} = p_{minor}$, $p_{0y} = p_{major} = 0.5 - p_{minor}$ for $\widetilde{P}$. For both $P^*$ and $\widetilde{P}$ we fix class marginals $p_{\cdot 0} = p_{\cdot 1} = 0.5$ and generate Gaussian features $X|A = a, Y = y \sim \mathcal{N}(\mu_{ay}, \Sigma_{ay})$ in 2-dimensions (see additional data generating details in Appendix D). In Figure 2 we show a qualitative example of simulated train data from $\widetilde{P}$ with $p_{minor} = 0.1$ and test data from $P^*$, and the corresponding decision boundaries of a baseline classifier and a classifier trained with the Equalized Odds fairness constraint (irregularities in the decision heatmaps are due to stochasticity in the Reductions prediction rule). In this example fair classifier is *3% more accurate*

on the test data and 1% less accurate on a biased test data sampled from $\widetilde{P}$ (latter not shown in the figure).

We proceed with a quantitative study by varying degree of bias in $\widetilde{P}$ via changing $p_{minor}$ in $[0.01, 0.25]$ and comparing performance of the baseline and fair classifier on test data from $P^*$ and $\widetilde{P}$. We present results over 100 runs of the experiment in Figure 3. Notice that the sum of each column of $\widetilde{P}_{A,Y} - P^*_{A,Y}$ is 0 for any value of $p_{minor}$ and we observe that the fair classifier has almost constant accuracy on $P^*$ (consistently outperforming the baseline), as predicted by Corollary 3.5. The largest bias in the training data corresponds to $p_{minor} = 0.01$, where baseline is erroneous on the whole $a = 1, y = 0$ subgroup (cf. Figure 2) resulting in close to 75% accuracy corresponding to the remaining 3 (out of 4) subgroups. For $p_{minor} = 0.05$ minority group acts as outliers causing additional

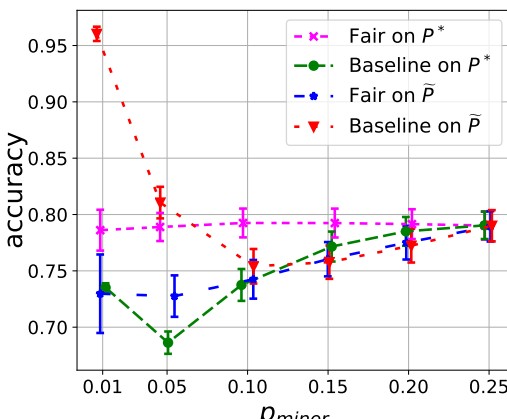

Figure 3: Test accuracy on $P^*$ and $\widetilde{P}$ when trained on the (biased) data from $\widetilde{P}$.

errors at test time resulting in the worst performance overall. When $p_{minor} = 0.25$, $\widetilde{P} = P^*$ and all methods perform the same as expected. Results on $\widetilde{P}$ correspond to the case where test data follows same distribution as train data, often assumed in the literature: here baseline can outperform fair classifier under the more extreme sampling bias conditions, i.e. $p_{minor} \leq 0.1$. We note that as the society moves towards eliminating injustice, we expect test data in practice to be closer to $P^*$ rather then replicating biases of the historical train data $\widetilde{P}$.

**Recidivism prediction on COMPAS data.**
We verify that our theoretical findings continue to apply on real data. We train baseline and fair classifier on COMPAS dataset (Angwin et al., 2016). There are two binary protected attributes, Gender (male and female) and Race

Table 1: Accuracy on COMPAS data

|  | Test on $P^*$ | Test on $\widetilde{P}$ |
|---|---|---|
| Fair | **0.652**±0.013 | 0.660±0.009 |
| Baseline | 0.634±0.011 | **0.668**±0.010 |

(white and non-white), resulting in 4 protected groups $A \in \{0, 1, 2, 3, 4\}$. The task is to predict if a defendant will re-offend, i.e. $Y \in \{0, 1\}$. We repeat the experiment 100 times, each time splitting the data into identically distributed 70-30 train-test split, i.e. $\widetilde{P}$ for train and test, and obtaining test set from $P^*$ by subsampling test data to preserve $Y$ marginals and enforcing equal representation at each of the 4 levels of the protected attribute $A$. We present results in Table 1. We see that our theory holds in practice: *accuracy of the fair classifier is 1.8% higher* on $P^*$. Baseline is expectedly more accurate on the biased test data from $\widetilde{P}$, but only by 0.8%.

We present results for the same experimental setup on the Adult dataset (Bache and Lichman, 2013) in Table 2 in Appendix D. We observe same pattern: in comparison to the baseline, fair classifier increases accuracy on $P^*$, but is slightly worse on the biased test data from $\widetilde{P}$.

## 5 SUMMARY AND DISCUSSION

We showed that enforcing algorithmic fairness allows practitioners to recover from certain biases in the training data. The main insight from our theoretical results is enforcing risk-based fairness constraints mitigates bias in the training data that is orthogonal to the fairness constraints. In other words, regardless of the magnitude of the training bias, the fairness constraints just "soaks it up". On the other hand, fairness constraints play no part in mitigating the remaining parts of the bias.

Our results depend on the assumption that the Bayes decision on test data satisfies a risk-based notion of algorithmic fairness. This assumption is strong, but it is necessary to recover the Bayes classifier. To remove this assumption, we must weaken the goal to merely improving upon the risk minimizer on the biased data. The set of $P^*$'s for which enforcing the fair constraint improves accuracy is easy to characterize. Let $\mathcal{F}$ be a fairness constraint and $\widetilde{R}$ and $\widetilde{R}_{\mathcal{F}}$ be the (unconstrained) and fairness

constrained risk minimizers with respect to $\widetilde{P}$:

$$\widetilde{R} \triangleq \arg\min_{R \in \mathcal{R}} \langle \widetilde{P}_{A,V}, R \rangle, \quad \widetilde{R}_{\mathcal{F}} \triangleq \arg\min_{R \in \mathcal{R} \cap \mathcal{F}} \langle \widetilde{P}_{A,V}, R \rangle.$$

The set of $P$'s for which enforcing the fairness constraint improves accuracy is $\{P : \langle P, \widetilde{R}_{\mathcal{F}} - \widetilde{R} \rangle \leq 0\}$. In other words, as long as $P^*$ is in the preceding set, then enforcing the fairness constraint improves accuracy. Studying the structure of this set is a promising area of future work.

We also note that there is another family of algorithmic fairness practices based on robust optimization (Hashimoto et al., 2018; Sagawa et al., 2019; Yurochkin et al., 2020) that are also widely used in practice. Although there are empirical results that demonstrate the efficacy of such practices, there are no theoretical results justifying their use when the training data is biased. This is another promising area of future work.

Taking a step back, the main takeaway for ML practitioners is possibility to encourage fairness and improve accuracy by enforcing risk-based fairness constraints during training. As long as they choose the discriminative attribute carefully so that the risk profiles are identical in the training data and at test time, then it is possible to learn the Bayes decision rule on the test data from (biased) training data. This departs from most prior work on algorithmic fairness that starts with the premise that fairness is an intrinsically desirable property of an ML model. Unfortunately, although most ML practitioners agree that algorithmic fairness is desirable, they are generally unwilling to sacrifice accuracy of the model for fairness. This is a gap between ML practice and algorithmic fairness research, and our work is one way to close this gap. By aligning algorithmic fairness with the usual goal of ML practitioners, we hope that this argument enlists the "invisible hand" of accuracy to promote algorithmic fairness practices.

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

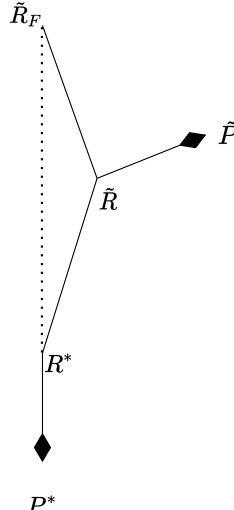

Figure 4: Example in which enforcing algorithmic fairness harms OOD generalization. The triangle is the set of risk profiles, and the dotted left side of the triangle intersects the fair constraint (*i.e.* the risk profiles on the dotted line are algorithmically fair). The training objective $\widetilde{P}$ is chosen so that the (unconstrained) risk minimizer on biased training data $\widetilde{R}$ is the vertex on the right and the fair risk minimizer (also on biased training data) $\widetilde{R}_{\mathcal{F}}$ is the vertext on top. The test objective points downward, so points close to the bottom of the triangle have the smallest risk at test time. We see that $\widetilde{R}$ is closer to the bottom of the triangle than $\widetilde{R}_{\mathcal{F}}$, so it has better OOD generalization.

## A  ENFORCING ALGORITHMIC FAIRNESS MAY HARD OOD GENERALIZATION

In this section, we provide an example in which enforcing algorithmic fairness harms OOD generalization performance. Formally, we provide choices of $\widetilde{P}$, $P^*$, $\mathcal{F}$, and $\mathcal{R}$ such that (i) $R^* \in \mathcal{F}$ and (ii) $\langle P^*, \widetilde{R} \rangle \leq \langle P^*, \widetilde{R}_{\mathcal{F}} \rangle$. Figre 4 shows our example. Inspecting the figure reveals such examples are hardly pathological. Just in this figure, there are a range of choice for $P^*$ and $\widetilde{P}$ that lead to enforcing algorithmic fairness harming OOD generalization.

## B  PROOF OF THEOREM 3.4

In this section, we provide a proof of Theorem 3.4 under the additional assumption that $\mathcal{A}$ and $\mathcal{V}$ are finite sets. Although less general, we feel that this proof is more instructive because it suggests the origin of (3.1).

*Proof.* **"if" direction:** Let $\widetilde{Z} = -\widetilde{P}_{A,V}$. If (3.1), then it is not hard to check that $(R^*, \widetilde{Z})$ satisfies the optimality conditions of (2.4):

$$
\begin{aligned}
0 &= \widetilde{P}_{A,V} + \widetilde{Z}, \quad \text{(stationarity)} \\
R^* &\in \mathcal{F}, \quad \text{(primal feasibility)} \\
\widetilde{Z} &\in \mathcal{N}_{\mathcal{R}}(R^*) + \mathcal{F}^{\perp} \quad \text{(dual feasibility)}.
\end{aligned}
\tag{B.1}
$$

Indeed, we have stationarity by the definition of $\widetilde{Z}$. We have primal feasibility because the unconstrained risk minimizer on unbiased data is algorithmically fair: $R^* \in \mathcal{F}$. We have dual feasibility because

$$
\begin{aligned}
\widetilde{Z} &= \Pi_{\mathcal{F}}(P^*_{A,V} - \widetilde{P}_{A,V}) - P^*_{A,V} + \Pi_{\mathcal{F}^{\perp}}(P^*_{A,V} - \widetilde{P}_{A,V}) \\
&\in \mathcal{N}_{\mathcal{R}}(R^*) + \mathcal{F}^{\perp} + \mathcal{F}^{\perp} \\
&= \mathcal{N}_{\mathcal{R}}(R^*) + \mathcal{F}^{\perp},
\end{aligned}
$$

where we appealed to (3.1) in the second step and recalled $\mathcal{F}^\perp$ is a subspace in the third step. The FRM problem (2.4) is convex, so (B.1) implies $R^*$ is an optimal point of (2.4).

**"only if" direction:** Assume $R^*$ solves (2.4). This implies there is $\widetilde{Z} \in \mathcal{N}_\mathcal{R}(R^*) + \mathcal{F}^\perp$ such that $(R^*, \widetilde{Z})$ satisfies (B.1). By the stationary and dual feasibility conditions,

$$\widetilde{Z} = -\widetilde{P}_{A,V} \in \mathcal{N}_\mathcal{R}(R^*) + \mathcal{F}^\perp.$$

We write $\widetilde{P}_{A,V}$ as $\Pi_\mathcal{F}(P^*_{A,V} - \widetilde{P}_{A,V}) - P^*_{A,V} + \Pi_{\mathcal{F}^\perp}(P^*_{A,V} - \widetilde{P}_{A,V})$ and rearrange to obtain

$$\Pi_\mathcal{F}(P^*_{A,V} - \widetilde{P}_{A,V}) - P^*_{A,V} \in \Pi_{\mathcal{F}^\perp}(P^*_{A,V} - \widetilde{P}_{A,V}) + \mathcal{N}_\mathcal{R}(R^*) + \mathcal{F}^\perp$$
$$= \mathcal{N}_\mathcal{R}(R^*) + \mathcal{F}^\perp,$$

where we recalled $\mathcal{F}$ is a subspace in the second step. $\qquad\square$

## C    CONTINUOUS DISCRIMINATIVE ATTRIBUTES

In this section, we state and prove a more general verion of Theorem 3.4 that permits continuous discriminative attributes. In this more general setting, risk profiles are (integrable) functions on $\mathcal{Z} \triangleq \mathcal{A} \times \mathcal{V}$, so the fair risk minimization problem (B.1) and its unconstrained counterpart are infinite dimensional optimization problems. We start by setting up the problem and reviewing relevant results from optimization theory.

Let $(\mathcal{Z}, \Sigma)$ be a measurable space and $\mathcal{S}$ be the set of bounded measurable functions on $(\mathcal{Z}, \Sigma)$. We equip $\mathcal{S}$ with the sup norm. The risk set $\mathcal{R}$ and the fair constraint set $\mathcal{F}$ are generally subsets of $\mathcal{S}$. The (topological) dual of $\mathcal{S}$ (denoted by $\mathcal{S}'$) is the set of finitely additive measures on equipped with the total variation norm Dunford et al. (1958). This result allows us to represent continuous linear functionals on such spaces with (finitely additive) measures, so it is a generalization of the more familiar Riesz–Markov–Kakutani representation theorem to spaces of (possibly discontinuous) measurable functions. We observe that the more familiar set of countably additive measures is a closed subset of $\mathcal{Z}'$.

**Definition C.1** (Complemented subspace). *Let $\mathbb{B}$ be a Banach space and $A \subset \mathbb{B}$ be a subspace. We say $A$ is complemented subspace of $\mathbb{B}$, if there exists another subspace $A_C \subset \mathbb{B}$ such that $\mathbb{B} = A \oplus A_C$.*

Henceforth, for if $A$ is a complemented subset of a Banach space $\mathbb{B}$, (*i.e.*, $A \oplus A_c = \mathbb{B}$) then we define $\Pi_{A,A_C}(x)$ (resp. $\Pi_{A_C,A}(x)$) is the component of $x$ in $A$ (resp. $A_C$), i.e. $\Pi_{A,A_C}(x) = x_1$ (resp. $\Pi_{A_C,A}(x) = x_2$) where $x = x_1 + x_2$ with $x_1 \in A, x_2 \in A_C$. Recall that, we define $\mathcal{F}$ as the fair hyperplane. Previously it was a subspace of the risk set, now it becomes a subspace of $\mathcal{S}$. We have the following assumption on the fair hyperplane:

**Definition C.2** (Annihilator). *For any $A \subset \mathbb{B}$ we define its annihilator $A^\perp \subset \mathbb{B}'$ as the set of bounded linear functions $f : \mathbb{B} \to \mathbb{R}$ such $f(x) = 0$ for all $x \in A$.*

**Lemma C.3.** *Let $A$ be a complemented subspace in $\mathbb{B}$. Then $A^\perp$ is complemented in $\mathbb{B}'$.*

*Proof.* Since, $A$ is complemented in $\mathbb{B}$, there exists a subspace $G \subset \mathbb{B}$ such that $A \oplus G = \mathbb{B}$. This implies, each $x \in \mathbb{B}$ has the unique decomposition $x = x_1 + x_2$, where $x_1 \in A$ and $x_2 \in G$. We consider the projection ma p $\Pi_{A,G} : \mathbb{B} \to \mathbb{B}$ such that $\Pi_{A,G}(x) = x_1$. Let us define two following subspaces in $\mathbb{B}'$ :

$$\mathcal{H}_{A,G} = \{f \circ \Pi_{A,G} \mid f \in \mathbb{B}'\}$$
$$\bar{\mathcal{H}}_{A,G} = \{f - f \circ \Pi_{A,G} \mid f \in \mathbb{B}'\}.$$

Note that, $\bar{\mathcal{H}}_{A,G} \subset A^\perp$. Also, for any $f \in A^\perp$ we have $f \circ \Pi_{A,G} = 0_{\mathbb{B}'} \implies f = f - f \circ \Pi_{A,G} \in \bar{\mathcal{H}}_{A,G}$. This implies $\bar{\mathcal{H}}_{A,G} = A^\perp$. Furthermore, $\mathbb{B}' = \mathcal{H}_{A,G} + \bar{\mathcal{H}}_{A,G}$ and for any $f \in \mathcal{H}_{A,G} \cap \bar{\mathcal{H}}_{A,G}$ we have $f(A) = f(G) = \{0\}$. Hence, $f = 0_{\mathbb{B}'}$. This implies $\mathbb{B}' = \mathcal{H}_{A,G} \oplus \bar{\mathcal{H}}_{A,G} = \mathcal{H}_{A,G} \oplus A^\perp$. $\qquad\square$

Finally, we review some relevant background on infinite dimensional optimization. Since we are mostly concerned with convex optimization problems with linear cost functions, the theory simplifies considerably.

**Definition C.4** (tangent cone). *The tangent cone of a closed convex set $\mathcal{C} \subset \mathbb{B}$ at a point $x \in \mathcal{C}$ is the closure of the cone of feasible directions at $x$:*

$$T_{\mathcal{C}}(x) \triangleq \mathsf{cl}\{d \in \mathbb{B} \mid \text{there is } \bar{t} > 0 \text{ such that } x + td \in \mathcal{C} \text{ for all } t \in [0, \bar{t}]\}.$$

There are many notions of tangent cone in variational analysis (*e.g.* Clarke tangent cone, contingent cone, inner tangent cone *etc.*), but they all coincide for closed convex sets Bonnans and Shapiro (2000). Notably, this definition is identical to the definition (for convex sets) in finite dimensions.

**Definition C.5** (normal cone). *The normal cone of a closed convex set $\mathcal{C} \subset \mathbb{B}$ at a point $x \in \mathcal{C}$ is the polar cone of the tangent cone of $\mathcal{C}$ at $x$:*

$$N_{\mathcal{C}}(x) \triangleq \{d' \in \mathbb{B}' \mid \langle d', d \rangle \leq 0 \text{ for all } d \in T_{\mathcal{C}}(x)\}.$$

**Proposition C.6.** *Let $\mathcal{C}$ be a closed convex subset of a Banach space $\mathbb{B}$. Consider the convex optimization problem*

$$\min_{x \in \mathcal{C}} \langle c, x \rangle.$$

*A point $x^* \in \mathcal{C}$ is an optimal point iff*

$$\langle c, d \rangle \geq 0 \text{ for any } d \in T_{\mathcal{C}}(x^*),$$

*where $\langle c, \cdot \rangle$ is the linear cost function and $T_{\mathcal{C}}(x^*)$ is the tangent cone of $\mathcal{C}$ at $x^*$. Equivalently, $x^*$ is optimal if and only if $c \in N_{\mathcal{C}}(x^*)$.*

Recall that in a normed vector space $\langle f, x \rangle$ means the value of the linear functional $f$ at $x$. In our problem setting, points in the normed space $\mathcal{S}$ are integrable functions/random variables and linear functionals on $\mathcal{S}$ are (finitely additive) measures, so $\langle f, x \rangle$ means expectation of the random variable x with respect to probability measure $f$.)

We are ready to state the extension of our main result to continuous discriminative attributes. Assumptions 3.1, 3.2, 3.3 from the main paper remain in effect. For continuous discriminative attributes, we impose an additional assumption.

**Assumption C.7.** *The fair subspace $\mathcal{F}$ is complemented in $\mathcal{S}$.*

This assumption is usually satisfied by common algorithmic fairness constraints: when RP is considered, $\mathcal{F}$ is the set of all constant functions from $\mathcal{A}$ to $\mathbb{R}$. For CPR, $\mathcal{F} \subseteq \mathcal{S}$ is the set of all functions $f : \mathcal{A} \times \mathcal{Y} \to \mathbb{R}$ such that $f$ is constant on the first co-ordinate, i.e. $f(x_1, y) = f(x_2, y)$ for all $x_1 \neq x_2 \in \mathcal{A}$ and $y \in \mathcal{Y}$. We now argue that, in both the cases $\mathcal{F}$ is a complemented subset of $\mathcal{S}$ under mild assumptions. For RP, we use the fact that any subspace $A \subseteq \mathcal{S}$ with $\dim(A) < \infty$ or $\text{codim}(A) < \infty$ is complemented. As $\mathcal{F}^{RP}$ is the set of all constant functions, it has dimension 1 and hence complemented. For CRP, assume that there exists some base measure $\mu$ such that $f \in \mathcal{S}$ is integrable with respect to $\mu$. Then one can write: $f = f_1 + f_2$ where $f_1 \in \mathcal{F}$ which is defined as: $f_1(a, v) = g(v)$ where $g(v)$ is the marginal of $f(\cdot, v)$ with respect to the base measure $\mu$. The function $f_2$ is analogously defined as $f - f_1 \equiv f(a, v) - g(v)$.

**Theorem C.8.** *If the unconstrained risk minimizer on unbiased data is algorithmically fair (i.e. its risk profile $R^*$ satisfy the fairness constraints), then fair risk minimization (2.4) learns $h \in \mathcal{H}$ such that $R(h) = R^*$ under assumptions C.7 and 3.2 if and only if*

$$\Pi_{\mathcal{F}_C^{\perp}, \mathcal{F}^{\perp}}(P_{A,V}^* - \widetilde{P}_{A,V}) - P_{A,V}^* \in \mathcal{N}_{\mathcal{R}}(R^*) + \mathcal{F}^{\perp}. \tag{C.1}$$

*where $P_{A,V}^*$ (resp. $\widetilde{P}_{A,V}$) is the marginal of $P^*$ (resp. $\widetilde{P}$) with respect to $(A, V)$, $\mathcal{N}_{\mathcal{R}}(R^*)$ is the normal cone of $\mathcal{R}$ at $R^*$ and $\Pi_{\mathcal{F}_C^{\perp}, \mathcal{F}^{\perp}}(\cdot)$ is the projection as defined previously.*

Table 2: Accuracy on Adult data

|      | $P^*$ | $\widetilde{P}$ |
|------|-------|-----------------|
| Fair | **0.852**±0.004 | 0.843±0.003 |
| Base | 0.848±0.005 | **0.847**±0.003 |

*Proof.* For notation simplicity define $X \triangleq \Pi_{\mathcal{F}_C^\perp, \mathcal{F}^\perp}(P_{A,V}^* - \widetilde{P}_{A,V}) - P_{A,V}^*$. We show that $\min_{R \in \mathcal{F}} \langle \widetilde{P}, R \rangle = \langle \tilde{P}, R^* \rangle$ holds if and only if $X \in \mathcal{N}_\mathcal{R}(R^*)$. Towards that end, fix $R \in \mathcal{F}$:

$$
\begin{aligned}
\langle \widetilde{P}, R \rangle &= \langle \widetilde{P} - P^*, R \rangle + \langle P^*, R \rangle \\
&= \langle \Pi_{\mathcal{F}_C^\perp, \mathcal{F}^\perp}(\widetilde{P} - P^*), R \rangle + \langle P^*, R \rangle \\
&= \langle -P^* - X, R \rangle + \langle P^*, R \rangle \\
&= \langle -X, R \rangle \qquad\qquad\qquad\qquad\qquad\qquad \text{(C.2)} \\
&= \langle -X, R^* \rangle + \langle -X, R - R^* \rangle \\
&= \langle \tilde{P}, R^* \rangle + \langle -X, R - R^* \rangle \quad \text{[From equation (C.2)]}
\end{aligned}
$$

Hence we have: $\min_{R \in \mathcal{F}} \langle \widetilde{P}, R \rangle = \langle \tilde{P}, R^* \rangle$ if and only if $\langle -X, R - R^* \rangle \geq 0$ for all $R \in \mathcal{F}$ which holds if and only if $X \in \mathcal{N}_{\mathcal{R} \cap \mathcal{F}}(R^*) = \mathcal{N}_\mathcal{R}(R^*) + \mathcal{F}^\perp$. This completes the proof. $\square$

## D   EXPERIMENTAL DETAILS

We provide additional details to help reproduce our results. Please also see the code provided with the submission. Code for the Reductions classifier (Agarwal et al., 2018) is available here: `https://github.com/fairlearn/fairlearn`. We modified the source code to prevent it from early stopping, so the baseline classifier runs for same number of iterations as the fair classifier. The idea behind the Reductions approach is to translate the problem of learning a fair classifier into a constraint optimization problem, where constraints depend on the fairness definition of choice. Reductions method requires a base classifier: it learns an ensemble of the base classifiers to optimize performance subject to the fairness constraints. We used logistic regression as the base classifier in all experiments. The other important parameter is the tolerance $\epsilon$ that controls the amount of permissible constraint violation. Smaller tolerance implies tighter fairness constraints. In all experiments we used Equalized Odds fairness constraint (Hardt et al., 2016) with $\epsilon = 10$ for the baseline classifier (i.e. fairness can be arbitrarily violated) and $\epsilon = 0.1$ (for the Adult experiment $\epsilon = 0.02$) for the fair classifier.[1]

**Simulations**   Simulated data is generated from $X|A = a, Y = y \sim \mathcal{N}(\mu_{ay}, \Sigma_{ay})$ in 2-dimensions with prescribed $A, Y$ joint distribution. We fixed $Y$ marginals $p_{\cdot 0} = p_{\cdot 1} = 0.5$ and varied joint $P_{A,Y}$ to study different degrees of label bias. Reductions was trained for 25 iterations for both baseline and fair classifiers. We provide code reproducing Figure 2 of the main text in `simulations.py`. Please also refer to the code for concrete values of $\{\mu_{ay}, \Sigma_{ay}\}$ and other minor details.

**COMPAS experiment**   Reductions was trained for 50 iterations for both baseline and fair classifiers. We provide code reproducing one run of the experiment for Table 1 of the main text (results in the table summarize 100 runs) in `compas.py`. Please also refer to the code for data pre-processing and other minor details.

**Adult experiment**   We ran experiment on the Adult dataset 2.

---

[1]We could use a simple logistic regression as the baseline classifier, however this would mean that baseline classifier and fair classifier are in different hypothesis classes. To avoid this, we used Reductions method for both with same number of iterations, however with loose fairness constraint for the baseline.

