# OpenReview forum: "There is no trade-off: enforcing fairness can improve accuracy"
_ICLR.cc/2021/Conference — Reject_

### Official Review · AnonReviewer4 · 2020-10-24
**Overly assumptions; no strong justifiable experimental results.**

**Rating:** 4
**Confidence:** 3

**Review:**

Although the paper studies a current and very important issue in algorithmic fairness, I have couple of concerns about the paper in general as follows:

1. The assumptions made in the paper makes the stated conclusion almost trivial making the technical contribution of the work not standing out. For instance, assumption 3.1 is not ever guaranteed to hold. Another major assumption is P* and the fact that the test data is coming from this distribution. This makes the problem trivial as it is almost saying that if I satisfy the fairness objective and make sure that my test data is satisfying the very same objective then the performance of the model will boost. This issue can also be observed in the experimental setting as described in my point #2 detailed below.

2. In experiments, the paper claims that we are forcing/incorporating the EO objective in our classification task but also making sure that the test data is satisfying the equal conditional factor. In this case yes the performance of the fair model on P* will increase, but as shown in the results, the performance of baseline model is better on P~ making the results not convincing to me.

3. I think there need to be more controlled and convincing experimental settings to show the significance of the claim. From the COMPAS results specifically, the baseline is clearly outperforming the fair classifier on P~. Yes sure the fair classifier will outperform on P* but this is only because P* is following and satisfying the objective of the fair classifier. This makes me wonder what the paper is trying to prove in this case.

4. Finally more datasets need to be tested specially to show the case. Having COMPAS only is not convincing in addition to the synthetic data which I noticed similar patter in it. The experimental setting is so brief specially on the real world benchmark fairness dataset, COMPAS.

5. A minor comment: The resume screening example brought up in the paper seems to me more like historical bias; however, authors refer to it as sampling bias.

6. Finally, the title makes a very strong claim which is not well supported in the paper considering both the experimental results and assumptions made in the paper.

Overall, the paper needs more experiments with clear and convincing setup with more benchmark datasets utilized in them. The paper can also improve its writing, clarity and accuracy of some minor points such as the sampling bias issue brought up in point 5.

---

> ### Author Response · Authors · 2020-11-24
> **Response to Reviewer 4**
>
> Please see our general response for the summary of the main changes. Specifically, we added a counterexample emphasizing that the intuition you mentioned "if I satisfy the fairness objective and make sure that my test data is satisfying the very same objective then the performance of the model will boost" is wrong. Our theoretical results show precisely when this is the case. We have also added an additional experiment on the Adult dataset (Table 2 in Appendix D) as you requested. We address your concerns below.
>
> **This makes the problem trivial as it is almost saying that if I satisfy the fairness objective and make sure that my test data is satisfying the very same objective then the performance of the model will boost.**
>
> Your intuition suggests enforcing algorithmic fairness always improves OOD generalization as long as the Bayes classifier at test time is algorithmically fair. This is incorrect. We provide a counterexample in Appendix A showing enforcing algorithmic fairness *can harm* OOD generalization *even if the Bayes classifier at test time is algorithmically fair*. We also refer to this example in the discussion of Assumption 3.1.
>
> **The performance of the fair model on P\* will increase, but as shown in the results, the performance of baseline model is better on P~ making the results not convincing to me. From the COMPAS results specifically, the baseline is clearly outperforming the fair classifier on P~.**
>
> We are not concerned with the performance of model on $\widetilde{P}$; we are concerned with their performance on $P^*$. This is the out-of-distribution (OOD) generalization problem, which we are casting the problem of algorithmic bias as an instance of. As ML methods being deployed in practice, it is evident that the performance on the iid test data ($\widetilde{P}$) is a poor indicator of OOD generalization. Topics such as adversarial robustness, group DRO, invariant risk minimization, and algorithmic fairness are just some of the examples of the iid assumption being challenged in the recent literature. In all of the aforementioned problems, performance on the iid test data is known to suffer to obtain practical generalization/robustness properties. Our setting is no-exception: fairness requires sacrificing some of the performance on the biased test data from $\widetilde P$.
>
> **Finally more datasets need to be tested specially to show the case. Having COMPAS only is not convincing in addition to the synthetic data which I noticed similar patter in it.**
>
> We would like to clarify that this is not a methodology paper. We are not proposing any new methods that need to be justified with many real data experiments. Our paper presents a theoretical study of existing algorithmic fairness practices, specifically focusing on when fairness can increase performance. In our experiments we use an existing fairness algorithm (Reductions) to illustrate and help the reader to understand the theory. That said, we have added an experiment on the Adult dataset following the setup of the COMPAS experiment in Table 2 in Appendix D.
>
> **The resume screening example brought up in the paper seems to me more like historical bias; however, authors refer to it as sampling bias.**
>
> Historical bias is not a statistical concept. It only means that some form of bias has occurred due to events in the past. In the resume screening example, historical bias is a sampling bias (a statistical concept).

---

### Official Review · AnonReviewer3 · 2020-10-28
**Nice result, more intuion/explaination is needed.**

**Rating:** 6
**Confidence:** 4

**Review:**


In this paper, the authors show that under some assumptions, they can use the fairness constraint to understand some parts of the distribution shift between train and test, and as a result they can improve accuracy at the test time. The main assumptions are (i) the optimal classifier satisfies the fairness constraint and (ii) the risk profiles of all the classifiers in their hypothesis class are the same between biased and unbiased distribution.

The paper was generally well-written; however, Theorem 3.4 and the assumption 3.3 (eq 2.3) were not sufficiently explained. For Assumption 3.3 the only apparent scenario is that if we group data points according to A and V then the weight of each group can vary between the train and test distribution. What other important distribution shift is supported under this assumption, and why are such distributions important/plausible? Also, the relation to the covariate shift requires some clarification.

As I explained, one of the supported distribution shifts is when groups (according to A, V) have different weights in train and test data. In this case, this work is similar to work on group DRO where they optimize the worst group risk instead of the average risk and show they can increase the accuracy in the test data. The authors need to compare their work with this line of work.

The intuition behind Theorem 3.1 (or at least what I understand) is that due to 3.2 we only need to consider the risk over groups according to A and V (RHS of 4.2). Depending on the weights of these groups, the optimal classifier is different.  The authors then assume in the test data the optimal classifier is fair; thus, this assumption leaks some information about the distribution of the test data (weights of these groups at the test data). By projecting the optimal classifier to the fair subspace, we can recover the optimal classifier under some assumption. Again, the only apparent scenario to me is when the distribution in test data has the same weights for the groups with different A but the same V. What other distributions can the test data have, in particular, for what distributions the fair classifier is also the optimal classifier?

I think Theorem 3.2 needs a better explanation. What is the intuition of the normal cone, in what conditions we cannot recover the optimal classifier, etc.? For example, although the paper said that showing 4 dimensions in figure 1 is hard, one can have 4 points with different risks on different groups and show which point is chosen by what algorithm and why.

If the optimal classifier satisfies some fairness constraint, then enforcing that fairness constraint on biased data should always increase the accuracy, right? So what is the surprising result here? Is it characterizing exactly when the optimal classifier is recovered?

Overall I think this was a good paper, and I enjoyed reading it. However, I believe more intuition and maybe more results will make the paper stronger.

---

> ### Author Response · Authors · 2020-11-24
> **Response to Reviewer 3**
>
> We thank you for the review. Please see our general response for the summary of the main changes. Specifically, we added a new example to section 2.2 (Example 2.6) that shows how the discriminative attribute can be used to broaden the applicability of our setting. We also added a counterexample showing that "if the optimal classifier satisfies some fairness constraint, then enforcing that fairness constraint on biased data should always increase the accuracy" is not always true. But, our theoretical results show precisely when this is the case. We have also revised the draft to improve clarity as you suggested. Please see our responses to the questions below.
>
> **For Assumption 3.3 the only apparent scenario is that if we group data points according to A and V then the weight of each group can vary between the train and test distribution. What other important distribution shift is supported under this assumption, and why are such distributions important/plausible? Also, the relation to the covariate shift requires some clarification.**
>
> First, we note that such marginal shifts assumptions are common in the literature on transfer learning. Given training examples in the form $(U,V)$, we can always factorize the joint distribution of the training examples as $P(U\mid V)P(V)$. Marginal shift means the marginal factor $P(V)$ is different in the source and target domains. By picking $U$ and $V$ carefully, it is not hard to see that label shift ($U = X$, $V = X$) and covariate shift ($U = Y$, $V = X$) are special instances of this assumption.
>
> Second, we remark that the choice of discriminative attribute makes this assumptions quite general. To highlight the generality of this assumption, we added another example to section 2.2. this new example uses the discriminative attribute in a crucial way.
>
> **Relation to DRO**
>
> Group DRO can be used for algorithmic fairness, but it is different enough from risk-based notions of algorithmic fairness that it deserves a separate investigation. We are not aware of any prior work deriving theoretical conditions under which group DRO improves test performance (as we do in our paper for fairness), but there are prior works demonstrating its utility empirically. We now mention group DRO in the discussion in Section 5.
>
> **The only apparent scenario to me is when the distribution in test data has the same weights for the groups with different A but the same V. What other distributions can the test data have, in particular, for what distributions the fair classifier is also the optimal classifier?**
>
> The condition that you have in mind is $P^* \in \mathcal F$, where $\mathcal F$ is the fair constraint set. While this is sufficient to ensure $R^* \in \mathcal F$, it is not necessary. If $\mathcal R$ is the convex hull of finitely many points, then all we need is $P^* \in N_{\mathcal R}(R^*)$, where $R^*$ is a point in $\mathcal F$. This is a much more general condition than $P^* \in \mathcal F$. For example, if different groups have different noise levels, then Assumption 3.1 can be satisfied for non-equal weights of the groups.
>
> **I think Theorem 3.2 needs a better explanation. What is the intuition of the normal cone, in what conditions we cannot recover the optimal classifier, etc.?**
>
> The normal cone captures the "natural" robustness of the optimality of $R^*$ to changes in $P^*$. In other words, how much an $P^*$ change, and the (unconstrained) risk minimization problem will still pick $R^*$ as the optimum. We clarify this in the discussion following Theorem 3.4 in the revised version of the paper.
>
> **If the optimal classifier satisfies some fairness constraint, then enforcing that fairness constraint on biased data should always increase the accuracy, right?**
>
> We provide a counterexample in Appendix A showing enforcing algorithmic fairness *can harm* OOD generalization *even if the Bayes classifier at test time is algorithmically fair*. We also refer to this example in the discussion of Assumption 3.1.

---

### Official Review · AnonReviewer2 · 2020-10-29
**There is no trade-off: enforcing fairness can improve accuracy**

**Rating:** 6
**Confidence:** 4

**Review:**

---------------------After reading the author response and the feedback from the others ------------------------

Thank the authors for their response.

I still think the assumptions made in the paper are strong.  For example, the assumption in equation 2.3 and 3.3 is hardly true. Even in the revised example 2.6, the missing rates might also depend on other factors. It is hard to identify every related attribute in general.

I am still not convinced that "machine-learning practitioners are unwilling to sacrifice the performance of their ML model for fairness". First, there might be many other possible reasons that some machine learning applications do not incorporate these fairness considerations. For example, there is no consensus on what fairness constraints we should achieve. In fact, existing works show that considering fairness might has bad effect on protected groups (e.g. Liu et al Delayed impact of fair machine learning). Machine-learning practitioners might just be confused what to do rather than unwilling to do. Second, I have seem many applications that have fairness incorporated. (e.g. Search Google image CEO, we now see many female CEOs on the top). From my perspective, responsible businesses would like to pay extra money to obtain a fair ML solution for the benefit of the society and the reputation of the business as long as they know what to do and how to achieve fairness.

So, I would like to keep my score.

------------------------------------Original Review---------------------------------------


Summary

The paper proposed risk parity and conditional risk parity as fairness measures that ensure the risk of machine learning models have the same risk for each subgroup. The paper discusses the conditions that minimizing the risk under a biased data distribution with these fairness constraints will lead to Bayes optimal policy in the target test distribution. Empirical evaluations on simulation data and rea-world recidivism prediction data validate the theoretical analysis.


Strengths:

1. The paper provides insight on when the optimal fair policy and the optimal utility policy coincide from a biased data point of view, i.e. ensuring some fairness on biased data might learn the optimal policy for the unbiased data. The discussion is very interesting.

2. The paper is clearly written and easy to follow.

3. The paper characterizes the necessary and sufficient for a policy that minimizes the risk under the proposed fairness constraints on the biased data distribution is optimal in the target test distribution.

Weakness:

1. Many assumptions are strong and not likely to be true in practice.

    1.1 For example, assumption in equation (2.3) or assumption (3.3) assumes that the risk of any policy on a subgroup is the same in both the biased data distribution and the true target test data distribution. In example 2.5, positive examples of disadvantaged group should be missing at random to satisfy the assumption. Missing at random is hardly true. For example, they might discard positive training examples based on some features, say the qualification of a female job candidate in the job screening example. Example 2.6 in general does not satisfy this assumption. I do not understand why the authors present this example 2.6.

    1.2 Assumption 3.1 assumes that the unconstrained risk minimizer on unbiased data distribution is algorithmically fair. This is hardly true.


2. If these assumptions do not hold, then the fairness definition sometimes does not make sense to me. For example, when assumption 3.1 is violated, the optimal classification accuracy for group A is 90%, the optimal classification accuracy fro group B is 85% and there is a policy \pi that achieves the optimal accuracy for both groups. But risk parity constraint says that, we cannot choose this optimal policy, we should select a policy that achieves 85% accuracy for both groups. This does not help group B but only harms group A.

3. It would be great to compare with methods that directly model the bias in the data. For example, using importance sampling to correct the distribution difference between the training data distribution and the target test data distribution.



Additional feedback

Contribution 1 "which which "-> "which"

The end of page 4 "that that" -> "that there"

It would be great to provide evidence that "machine-learning practitioners are unwilling to sacrifice the performance of their ML model for fairness" in the abstract and in the conclusion.

---

> ### Author Response · Authors · 2020-11-24
> **Response to Reviewer 2 [Part 1]**
>
> We thank you for the feedback. Please see our general response for the summary of the main changes. We have fixed the typos you mentioned. Please see our response to your comments below:
>
> **For example, assumption in equation (2.3) or assumption (3.3) assumes that the risk of any policy on a subgroup is the same in both the biased data distribution and the true target test data distribution. In example 2.5, positive examples of disadvantaged group should be missing at random to satisfy the assumption. Missing at random is hardly true. For example, they might discard positive training examples based on some features, say the qualification of a female job candidate in the job screening example. Example 2.6 in general does not satisfy this assumption.**
>
> We would like to clarify that our assumptions permit biases significantly more general than missing at random, and that Example 2.6 does satisfy the assumptions we make. Eq. (2.3) and Assumption 3.3 are for risks **conditioned** on the protected attribute $A$ and another arbitrary attribute $V$. In the scenario you mentioned, the missing rate of female applicants depends on their qualifications. Then picking the discriminative attribute $V$ to be some measure of their qualifications allows the setup to include this type of sampling bias. We added this scenario as Example 2.6 in a revised version of the paper.
>
> **Assumption 3.1 assumes that the unconstrained risk minimizer on unbiased data distribution is algorithmically fair. This is hardly true.**
>
> This is an implicit assumption in many works that suggests collecting representative training data as a remedy for algorithmic bias. Such suggestions implicitly assume the Bayes classifier on representative training data is algorithmically fair. We elaborate on this in the paragraph following the statement of Assumption 3.1.
>
> **If these assumptions do not hold, then the fairness definition sometimes does not make sense to me. For example, when assumption 3.1 is violated, the optimal classification accuracy for group A is 90\%, the optimal classification accuracy for group B is 85\% and there is a policy $\pi$ that achieves the optimal accuracy for both groups. But risk parity constraint says that, we cannot choose this optimal policy, we should select a policy that achieves 85\% accuracy for both groups.**
>
> We wish to point out that risk parity is merely an instance of the risk-based algorithmic fairness definitions. In the example you mentioned, it does not make sense to use risk parity, but it may make sense to enforce a different risk-based algorithmic fairness (with a non-trivial discriminative attribute). We also note that such examples can be constructed to criticize any of the group fairness definitions. Specifically in the example you mentioned, definitions such as *error rate balance* (Chouldechova, 2017) and *overall accuracy equality* (Berk et al., 2017) will also lead to reduced performance on group A. Formulating "no-harm" fairness definitions is an interesting problem, but is beyond the scope of our work.
>
> Taking a step back, risk-based notions of algorithmic fairness are widely used in practice to the extent that there are special software packages developed to enforce such notions during training (e.g. TensorFlow Constrained Optimization). Our point is they implicitly come with this assumption of identical risk profiles in the source and target (otherwise enforcing risk-based notions of fairness during training may not translate to fairness in the target domain).

---

> > ### Author Response · Authors · 2020-11-24
> > **Response to Reviewer 2 [Part 2]**
> >
> > **It would be great to compare with methods that directly model the bias in the data. For example, using importance sampling to correct the distribution difference between the training data distribution and the target test data distribution.**
> >
> > Consider risk parity (Definition 2.1): to use importance sampling one must know marginal distribution of the protected attribute $A$ in $P^*$ (unbiased test distribution). Our results are more general: we show that training on $\widetilde P$ under the risk parity constraint will perform well on $P^*$ regardless of the $A$ marginal on $P^*$. The use case for importance sampling and fair risk minimization are different. If you know the distribution of the population segments at test time, then it is possible to use importance sampling to reweight the training data so the empirical risk minimization objective (during training) is unbiased for the risk at test time. Fair risk minimization leverages a different form of knowledge about the data at test time; instead of knowledge of the distribution of population segments, fair risk minimization leverages knowledge that the Bayes decision rule is fair.
> >
> > **It would be great to provide evidence that "machine-learning practitioners are unwilling to sacrifice the performance of their ML model for fairness" in the abstract and in the conclusion.**
> >
> > Our motivation for this study is the lack of broader adoption of the myriad of algorithmic fairness practices that have been developed over the past few years by the research community. Despite the availability of methods and implementations, this lack of adoption by practitioners suggests their incentives are misaligned with those of algorithmic fairness research. This study is an attempt at aligning these goals.
> >
> > It is honestly hard to imagine a business paying extra money to obtain a fair ML solution, especially when they already have one satisfying their business needs. It is even harder when the business believes that the fair ML solution will hurt their revenue. We can also consider publicly available pre-trained models, e.g. GPT, BERT, word embeddings, ImageNet models, which were all trained without fairness considerations.

---

### Author Response · Authors · 2020-11-24
**Summary of the main changes**

We thank the reviewers for their feedback. The main changes in the revised version are

* an additional example (Ex 2.6) demonstrating the generality of the assumptions on the bias in the training data;
* additional explanations and justifications for the assumptions;
* a counterexample in Appendix A showing that if the optimal classifier satisfies some fairness constraint, then enforcing that fairness constraint on biased data **may hurt** the accuracy.

---

### Decision · Program_Chairs · 2021-01-07
**Final Decision**

**Decision:**

Reject

**Comment:**

The problem as formalized in this paper is essentially a domain adaptation problem. There is a training distrinution P and a test distribution P*. the learner gets training data generated by P and aims to minimize the loss of its hypothesis w.r.t. P*. How is it relate dto fairness? The authors add the assumption "we assume the unbiased Bayes decision rule is algorithmically fair in some sense and hope that enforcing the correct notion of fairness allows us to recover h∗ from P". Under such an assumption, almost by definition "enforcing fairness may improve accuracy". By a similar logic, if we assume the unbiased byes decision rule is biased against a certain group, then enforcing bias against that group will imporve accuracy ....